# Do GLP-1 Analogs Have a Place in the Treatment of PCOS? New Insights and Promising Therapies

**DOI:** 10.3390/jcm12185915

**Published:** 2023-09-12

**Authors:** Aleksandra Szczesnowicz, Anna Szeliga, Olga Niwczyk, Gregory Bala, Blazej Meczekalski

**Affiliations:** 1Department of Gynecological Endocrinology, Poznan University of Medical Sciences, 61-701 Poznan, Poland; ola.szczesnowicz@gmail.com (A.S.); annamaria.szeliga@gmail.com (A.S.); olga.niwczyk@gmail.com (O.N.); 2UCD School of Medicine, University College Dublin, D04 V1W8 Dublin, Ireland; greg.bala1@gmail.com

**Keywords:** polycystic ovary syndrome, glucagon-like-peptide receptor agonists, metabolic syndrome, obesity, insulin resistance, liraglutide, semaglutide

## Abstract

Polycystic ovary syndrome (PCOS) is the most prevalent endocrinopathy in women of reproductive age. This condition is characterized by hyperandrogenism and either oligo- or anovulation. PCOS patients often present comorbidities such as obesity, insulin resistance, impaired glucose metabolism, dyslipidemia, hypertension, metabolic syndrome, and an increased risk of diabetes. Given the profound implications of metabolic impairment in PCOS, the accurate diagnosis and management of these facets are imperative. The first-line approach to treatment involves lifestyle modifications, including dietary adjustments and exercise aimed at achieving weight loss, a strategy consistently emphasized across the literature. Supplementation with probiotics, vitamin D, and L-carnitine have also provided additional benefits to patients. In select cases, pharmacological interventions are needed for optimal therapeutic results. The most common medications used in PCOS include metformin, thiazolidinediones, inositols, and two classes of antidiabetic agents: dipeptidyl peptidase-IV (DPP-IV) inhibitors, and sodium–glucose cotransporter-2 (SGLT-2) inhibitors. Glucagon-like peptide-1 receptor agonists (GLP-1RAs) are a new addition to the therapeutic arsenal for the metabolic management of PCOS. GLP-1 receptor agonists cause insulin release in a glucose-dependent manner, yielding clinical benefits such as heightened satiety, reduced appetite, and appetite regulation. GLP-1RAs have demonstrated efficacy in reducing glycated hemoglobin levels and promoting weight loss while ameliorating hyperlipidemia. Prior to initiating GLP-1RA therapy, patients should undergo screening for contraindications, including history of pancreatitis, diabetic retinopathy, or thyroid cancer. The effects of treatment should be monitored using laboratory testing and body weight measurements. Effective communication between clinician and patient should be maintained with regular check-in for a period of 6 to 12 months.

## 1. PCOS—A Short Overview of Metabolic Problems

Polycystic ovary syndrome (PCOS) is the most prevalent endocrinopathy among women of reproductive age. Its prevalence is estimated to range from 6% to 20% [1] within this demographic. The syndrome is typified by hyperandrogenism and ovulation dysfunction (oligo and/or anovulation).

The diagnostic criteria for PCOS were established two decades ago, precisely in 2003, as the Rotterdam criteria [2]. Notably, while these criteria do not encompass metabolic facets, metabolic challenges hold pivotal clinical significance for PCOS patients. These challenges have the potential to interfere with the menstrual cycle, the ovulatory process, and hyperandrogenism, consequently exerting adverse effects on long-term health outcomes.

PCOS patients commonly exhibit an elevated BMI and waist circumference, alongside insulin resistance, impaired glucose metabolism, dyslipidemia, hypertension, and metabolic syndrome. This spectrum of issues is pertinent to a wide range of PCOS patients, including adolescents, women in their reproductive years, pregnant women, and postmenopausal women [3].

Obesity is one of the most important metabolic issues afflicting PCOS patients, with estimates suggesting that at least 50% of individuals with PCOS are obese [4]. The incidence of obesity among PCOS patients is three-fold higher compared with women without PCOS [5]. Moreover, the occurrence of metabolic syndrome in PCOS patients is itself a multifaceted and intricate issue. Patients with PCOS in their reproductive years exhibit a two- to three-fold increased risk of metabolic syndrome compared with their healthy counterparts [4].

Insulin resistance (IR) is a hallmark of PCOS, with approximately 50–80% of PCOS patients affected [6]. The underlying mechanisms responsible for this dysfunction within PCOS are diverse. The most widely understood mechanism involves insulin receptor abnormality. In these cases, IR develops subsequently into a binding defect caused by excessive serine phosphorylation and diminished tyrosine phosphorylation [7]. Alternative mechanisms are linked to microRNA alterations that are observed in PCOS patients and suspected mitochondrial dysfunction linked to oxidative stress contributing to IR [8]. Furthermore, disturbances in normal intestinal flora have also been shown to play a role in the pathomechanism of IR [9].

The risk of developing type 2 diabetes is significantly increased, approximately threefold, in individuals affected by PCOS. Among those with PCOS and a family history of type 2 diabetes, there is a potential for the development of impaired insulin secretion and resistance [10]. Notably, PCOS patients exhibit distinctive atherogenic lipid profiles, characterized by elevated serum levels of LDL-cholesterol, decreased levels of HDL-cholesterol, and an increase in serum triglyceride levels [11].

PCOS patients can present with a range of cardiovascular dysfunctions, including hypertension, atherosclerosis, and coronary artery disease. Markers of these dysfunctions often correlate with flow-mediated dilatation, carotid intima-media thickness, and the presence of coronary artery disease [12]. A novel facet of the metabolic spectrum within PCOS is nonalcoholic fatty liver disease (NAFLD). Elevated serum androgen levels are contributory to the development of NAFLD in PCOS patients. Moreover, mitochondrial dysfunction is also considered a potential contributor to the increased prevalence of NAFLD in PCOS [13].

Sleep apnea is yet another condition associated with PCOS and of which PCOS patients have an increased risk of developing [14].

Given these complexities, it is imperative to ensure accurate diagnosis of metabolic impairments in PCOS patients. Furthermore, those identified with metabolic issues among PCOS patients should receive appropriate treatment using both non-pharmacological and pharmacological therapy.

## 2. GLP-1 Agonists: Overview, Classification, Pharmacological Applications, and Indications

Glucagon-like peptide-1 (GLP-1) agonists, also referred to as incretin mimetics or GLP-1 analogs, constitute a class of medications initially used in the management of type 2 diabetes mellitus (T2DM). Currently, a lot of attention is being directed toward the utilization of GLP-1 analogs in the treatment of obesity. Furthermore, ongoing investigations are exploring their potential utility for various other indications, including cardiovascular risk reduction in T2DM, treatment of nonalcoholic steatohepatitis (NASH), and Alzheimer’s disease. 

GLP-1, an incretin hormone, exhibits the capacity to stimulate insulin secretion in a glucose-dependent manner. The principal effect of GLP-1 analogs is similar to that of native GLP-1, as all synthetic GLP-1 receptor agonists bind to the GLP-1 receptor, eliciting glucose-dependent insulin release from pancreatic islets. Additional benefits of GLP-1 analogs include delayed gastric emptying and the inhibition of glucagon production by pancreatic alpha cells. Furthermore, patients who were started on a GLP-1 analog observed an average weight loss of 2.9 kg compared with the placebo. 

Coveleski et al. conducted a study showcasing that the administration of GLP-1 analogs reduced the sensation of hunger in obese women through direct enhancement in functional connectivity between the left nucleus tractus solitarius and the left thalamus and hypothalamus [15]. These analogs can also act indirectly by inhibiting neuropeptide Y and agouti-related peptides, resulting in heightened satiety and reduced hunger [16].

Structurally, two categories of GLP-1 agonists exist: those with a human GLP-1 backbone and those with an exendin-4 backbone. The former includes dulaglutide, abiglutide, liraglutide, semaglutide, while the latter include exenatide and lixisenatide [16]. While initial GLP-1 agonists were available solely in injectable formulations, recent developments have introduced oral formulations, such as oral semaglutide [17].

The indications for the introduction of GLP-1 analogs to treatment include a BMI of ≥30 kg/m^2^ or a BMI of ≥27 kg/m^2^ and at least one weight-related comorbidity including hypertension, dyslipidemia, type 2 DM, or obstructive sleep apnea [18].

The integration of GLP-1 analogs into therapy has been shown to contribute to an overall decreased risk of hypoglycemic events. When combined with basal insulin, GLP-1 receptor agonists exhibit potent glucose-lowering effects while minimizing weight gain and hypoglycemia when compared with intensified insulin regimens [16].

For the treatment of obesity, the subcutaneous administration of semaglutide and liraglutide has received approval for the treatment of obesity with or without underlying diabetes. Notably, semaglutide is the preferred therapeutic option in these cases as it has superior efficacy for weight loss over liraglutide. A study by Rubino et al. demonstrated that weekly subcutaneous semaglutide led to significantly greater weight loss than daily subcutaneous liraglutide among overweight and obese adults without diabetes [19].

However, the long-term safety of GLP-1 receptor agonists remains to be fully established. The side effects of GLP-1-based therapies are predominantly gastrointestinal, including nausea, vomiting, and diarrhea. An analysis of potential safety and tolerability issues associated with GLP-1 analogs found the risk of hypoglycemia, injection-site reactions, pancreatitis, neoplasms, and gallbladder events to be generally low [20]. Moreover, preclinical studies have suggested a potential link between GLP-1 analogs and an increased risk of thyroid C-cell tumors in rodents [21].

A comparison among GLP-1 agonists with respect to routes of administration, dosage, half-life time, and health effects is presented in Table 1.

## 3. Utilization of GLP-1 Analogs in the Context of Obesity

GLP-1 analogs possess the capability to induce hypoglycemic effects and facilitate weight reduction. Numerous clinical trials have substantiated the efficacy and safety of GLP-1 receptor agonists for the treatment and prevention of obesity [22]. These analogs have been identified as instrumental in transmitting meal-associated hunger and satiety signals to the brain [23].

GLP-1 analogs promote weight loss via numerous mechanistic routes [23], including:Stimulation of insulin production;Suppression of glucagon secretion;Delaying gastric emptying;Satiety enhancement;Hunger reduction;Decreasing energy intake;Regulation of appetite and food reward.

The intricate interplay between brain and gut hormones orchestrates the roles of both physiological and pharmacological GLP-1 in modulating appetite [24]. Multiple studies have addressed the effects of GLP-1 analogs on appetite, food preferences, and gastrointestinal hormones using varied doses, duration, and assessment methodologies [25,26].

In a three-year trial focused on obesity and prediabetes, it was found that liraglutide at a dosage of 3mg exhibited potential health benefits by reducing diabetes risk among patients with obesity and prediabetes [22]. In another investigation led by le Roux C. et al., liraglutide facilitated greater weight loss compared with the placebo at week 160 during the treatment period (liraglutide: –6.1% (SD 7.3) vs. placebo: −1.9% (SD 6.3); estimated treatment difference: −4.3%, 95% CI −4.9 to −3.7, *p* < 0.0001). The weight loss achieved with liraglutide was sustained for over three years. While some weight rebound was observed in the liraglutide group after treatment cessation at week 160, the treatment-associated difference remained significant at week 172 (–3.2%, 95% CI −4.3 to −2.2, *p* < 0.0001) [27].

GLP-1 contributes to the delay of gastric emptying and modulation of gut motility in healthy individuals, both lean and obese, as well as those with T2DM [24]. Furthermore, it significantly inhibits gastric emptying, preventing swift glucose entry into systemic circulation—a pivotal factor in postprandial glucose fluctuations [28]. Additionally, GLP-1 plays a role in the adjustment of gastric volume in anticipation of food intake, known as gastric accommodation. This process can influence the perception of stomach distention after consuming food [24].

In the context of glycemic control, GLP-1 analogs exhibit considerable potential in reducing glycated hemoglobin and fasting blood glucose levels [22]. Their chief known functions are linked to glucose metabolism, increasing pancreatic ß-cell insulin secretion (in a glucose-dependent manner), and the inhibition of hepatic glucose production via reduced α-cell glucagon secretion [28]. Notably, animal studies have suggested that GLP-1 may increase ß-cell mass through regeneration and a reduction in apoptosis [28,29].

The activation of GLP-1 receptors plays a dual role not only in simulating postprandial insulin secretion but also in extrapancreatic effects, both peripherally and centrally. It is via these pathways that they contribute to weight reduction by promoting satiety and delaying gastric emptying [30]. Moreover, exogenous GLP-1 analogs exert their effect on weight loss in a dose-dependent manner and in step with a reduction in appetite scores and caloric consumption. This establishes GLP-1 analog therapy as a compelling and attractive option in the management of obesity [23].

In a 2020 study, thirty-five participants in a liraglutide group demonstrated significant reductions in prospective food consumption scores and in the desire for sweet, salty, savory, or fatty foods, as well as heightened perception of fullness [26]. The physiological significance of endogenous GLP-1 in regulating appetite and eating behavior is underscored by data showing that the systemic administration of GLP-1 in humans reduces food intake by augmenting satiety and reducing the impetus to eat [28]. Notably, a study conducted by Flint A et al. revealed that GLP-1 infusion substantially impacted sensations of satiety (*p* = 0.013), hunger (*p* = 0.012), fullness (*p* = 0.028), and prospective food consumption (*p* = 0.012). Subjects reported increased satisfaction, fullness, and reduced hunger leading, to lowered anticipated food intake compared with patients receiving saline infusion. However, subjective ratings of taste, visual appeal, smell, aftertaste, and overall palatability of test meals did not differ on two experimental days [31].

In one study, the side effects of the administration of liraglutide were primarily associated with gastrointestinal symptoms. Among patients, 41% experienced nausea, 25% encountered diarrhea, 22% reported constipation, 20% had vomiting, and 10% had dyspepsia [27]. Additionally, headache (18%), back pain (13%), nasopharyngitis (26%), and upper respiratory tract infection (16%) were also reported [27]. Another study led by Bunck M. et al. also outlined mild to moderately intense gastrointestinal symptoms, including nausea (38.1%), vomiting (9.5%), abdominal distention (4.8%), and diarrhea (4.8%) [32]. Nonetheless, it is important to note that the treatment of obesity with GLP-1 analogs often necessitates higher doses compared to the treatment for T2DM. Consequently, the potential for gastrointestinal side effects can potentially be amplified in this group of patients [33].

While there have been isolated reports of acute pancreatitis among patients treated with GLP-1 analogs, extensive studies assessing the effects of these analogs have not demonstrated an increased risk of pancreatitis. Nonetheless, vigilant monitoring for signs and symptoms of acute pancreatitis remains advisable [33].

## 4. Management of Metabolic Issues in PCOS

The improvement in hyperandrogenism, reproductive function, and metabolic parameters including hyperlipidemia, glycemic control, and hypertension in women with PCOS has been demonstrated with weight reduction [18]. The current first-line approach to treatment involves lifestyle interventions including diet, exercise, and behavioral therapy [34].

Physical exercise plays a vital role in normalizing metabolic parameters. PCOS guidelines advocate for a minimum of 150 min of physical activity per week [35]. Moderate aerobic exercise has been shown to potentiate insulin sensitivity in PCOS. Studies have indicated that vigorous aerobic exercise and resistance training can lead to improvements in insulin sensitivity and abnormal androgen levels in women with PCOS. To maximize efficacy, a weekly regimen should include a minimum of 90 min of intensive exercise [36].

Dietary interventions hold therapeutic promise for various chronic diseases, particularly those linked to metabolic syndrome, by modulating the microbiome [37]. In the context of PCOS, dietary interventions are considered safe and convenient long-term treatment options. A prolonged intake of a low carbohydrate diet (LCD) has demonstrated beneficial effects on PCOS. Following a ketogenic diet has also been suggested to alleviate symptoms such as irregular menses and impaired liver function associated with PCOS [10].

Furthermore, following a reduced-calorie diet with a low glycemic index (GI) can prove beneficial for PCOS patients struggling with metabolic disturbances. Such diets have shown decreased homeostatic model assessment for insulin resistance (HOMA-IR), fasting insulin levels, total and low-density lipoprotein (LDL) cholesterol, triglycerides, waist circumference, and total testosterone [38].

However, it is important to note that patients frequently exhibit resistance toward therapeutic lifestyle interventions, resulting in low adherence and diminished treatment efficacy. Dropouts are common, as shown in a lifestyle intervention study by Fong et al., who observed discontinuation rates varying between 12% and 47% in overweight and obese women with PCOS [39].

In recent years, the relationship between PCOS and changes in gut microbiota has been extensively investigated. A significant difference in the composition of the gut microbiome has been observed between patients with PCOS and healthy controls [39]. A study by Singh et al. observed that animal-based proteins were associated with elevated levels of *Bacteroides*, *Alistipes,* and *Bilophila*, as well as a reduced count of *Bifidobacterium (B.) adolescentis*, a profile associated with an increased risk of cardiovascular disease [40].

In women with PCOS, inflammation underlies pancreatic beta-cell dysfunction, insulin resistance, and atherogenesis. Research indicates that the concurrent administration of a probiotic mixture (comprising *Lactobacillus acidophilus*, *Lactobacillus reuteri*, *Lactobacillus fermentum*, and *Bifidobacterium bifidum*) and a daily 200 μg dose of selenium for 12 weeks yields favorable outcomes in mental health parameters, serum total testosterone, hirsutism, hs-CRP, total antioxidant capacity, and malondialdehyde levels in women with PCOS [41].

Beyond probiotic supplementation, the intake of vitamin D and L-carnitine has also demonstrated benefits for PCOS patients. Vitamin D contributes to insulin synthesis, enhances insulin receptor expression, and amplifies insulin response to glucose transport [42]. Studies have indicated that women with PCOS receiving a weekly dose of 20,000 IU of cholecalciferol experienced enhanced carbohydrate metabolism, including a reduction in fasting glucose, triglycerides, and estradiol levels [43].

Studies have highlighted the potential of L-carnitine to reduce insulin levels and elevate serum adiponectin. Notably, treatment has been associated with a decline in HOMA-IR. Significant improvements in testosterone, FSH, and LH levels were also reported [44]. However, a 12-week trial demonstrated that L-carnitine exhibited no discernable effect on SHBG and lipid profile [45].

Complementing lifestyle interventions and dietary modifications, pharmacological interventions also play a pivotal role in the metabolic management of PCOS. Among the most commonly used medications are oral insulin sensitizers, which include metformin, thiazolidinediones, inositols, and berberine. Additionally, two categories of antidiabetic drugs, namely, dipeptidyl peptidase-4 (DPP-4) inhibitors and sodium–glucose co-transporter-2 (SGLT2) inhibitors, are also used in the treatment of PCOS.

Metformin, which lowers serum glucose levels by inhibiting hepatic glucose production [46], is recommended as a first-line medication for women with PCOS and T2DM or impaired glucose tolerance and have not responded to lifestyle modification. In women with PCOS exhibiting menstrual irregularity where hormonal contraceptives are either contraindicated or not tolerated, metformin is advised as second-line therapy [47].

Treatment with metformin significantly improves body mass index (BMI), serum lipids, and glucose homeostasis. Combining lifestyle adjustments and metformin yields superior results compared with lifestyle modification alone, particularly in reducing BMI and adipose tissue levels [48].

Thiazolidinediones mitigate insulin resistance through activation of the PPARγ receptor and have been found to improve hyperglycemia and dyslipidemia. Additionally, they improve menstrual cycle irregularity, promote ovulation, and reduce androgen levels in women with PCOS [49,50]. However, their utilization is constrained due to their numerous side effects such as weight gain, peripheral edema, and heart failure [48].

Inositols act as secondary messengers for insulin, mediating its diverse functions. Myo-inositol (MI) and D-chiro-inositol (DCI) are two isomers used in the treatment of PCOS. DCI restores insulin sensitivity by curtailing circulating androgens, while MI primarily exerts its benefits at the ovary level, where it is highly concentrated, aiding follicle-stimulating hormone (FSH) signaling. Both MI and DCI effectively reduce luteinizing hormone (LH) levels, LH/FSH ratios, and testosterone levels in PCOS. Simultaneous administration of MI and DCI in a 40:1 ratio has shown efficacy in restoring ovulation [51].

DPP-4 inhibitors increase the active levels of incretin hormones in the body, regulating elevated blood glucose by stimulating pancreatic insulin secretion, suppressing pancreatic glucagon secretion, and signaling the liver to reduce glucose production [52]. Notably, sitagliptin, alogliptin, and linagliptin have been studied in PCOS, revealing that treatment with sitagliptin at 100mg/day for 1 month effectively reduces blood glucose and visceral fat while also improving beta-cell function and preventing impaired glucose tolerance [51].

SGLT2 inhibitors decrease blood glucose levels by promoting glucosuria and natriuresis. These inhibitors target glucose reabsorption from the proximal tubules of the kidneys [53]. Empagliflozin, in comparison with metformin, demonstrated significant improvements in weight loss, BMI, waist-to-hip ratio, and fat mass reduction [51]. Furthermore, 50mg of licogliflozin administered three times daily effectively reduced hyperinsulinemia and hyperandrogenemia in women with PCOS, hinting at its potential as a promising novel treatment [54].

## 5. Potential Role of GLP-1 Analogs in PCOS: Indications, Patient Qualification, and Monitoring

GLP-1 receptor agonists (GLP-1RAs) such as liraglutide, exenatide, and semaglutide have emerged as novel therapeutic prospects for PCOS, owing to their distinct advantages in treating metabolic disorders. Longer-acting GLP-1RAs and once-weekly formulations have demonstrated superior glucose-lowering potential with less gastrointestinal discomfort compared with shorter-acting counterparts. Greater weight loss, particularly at higher doses of semaglutide and dulaglutide, has also been shown [55].

In diabetic patients, the use of GLP-1RAs has been associated with significant reductions in glycated hemoglobin, weight loss, modest lowering of blood pressure, and improved hyperlipidemia [56,57]. A study by Astrup et al., comparing the efficacy of GLP-1 agonists versus orlistat in treating obesity, demonstrated that liraglutide 3 mg led to a 76% weight loss exceeding 5%, outperforming both the placebo (30%) or orlistat (44%). Liraglutide across doses also reduced blood pressure and markedly reduced the prevalence of prediabetes (84–96% reduction) at doses of 1.8 to 3.0 mg per day [58].

Incorporating women with PCOS and obesity who were previously treated with metformin, a randomized trial conducted by Jensterle et al. reported greater BMI reductions with daily administration of 1.2 mg liraglutide compared with 1000 mg metformin twice daily (reductions of 1.1 ± 1.26 kg/m^2^ for liraglutide versus 0.1 ± 0.67 kg/m^2^ for metformin). Liraglutide treatment also significantly reduced visceral adipose tissue area [59].

Subcutaneous liraglutide 3 mg once daily is indicated as an adjunct to chronic body weight management in adults with a BMI ≥30 kg/m^2^ or BMI of ≥27 kg/m^2^ and at least one weight-related comorbidity, including hypertension, dyslipidemia, obstructive sleep apnea, or type 2 diabetes [18]. A trial by Frøssing et al., investigating 72 overweight women with PCOS who were treated with liraglutide 1.8 mg/day or placebo for 26 weeks showed that treatment with liraglutide reduced body weight significantly by over 5%, liver fat by 44%, visceral adipose tissue by 18%, and free testosterone level by 19% [60].

The anti-inflammatory effect of GLP-1RAs is not limited to adipose tissue. PCOS and IR are risk factors for damage to other tissues, including the vascular endothelium.

A study investigating the impact of liraglutide treatment on atherothrombotic risk was performed which included 19 obese women with PCOS and a control group of 17 healthy people. At six months, the liraglutide group had experienced weight loss (3–4%) and had decreased atherothrombosis markers such as inflammation, endothelial function, and clotting [61].

In a study by Elkind-Hirsch et al., the efficacy of liraglutide was tested against placebo in women with PCOS. The authors found that liraglutide at a dose of 3 mg once daily was superior to placebo for weight loss, reducing hyperandrogenism, and improving cardiometabolic parameters among women with PCOS and obesity [62].

GLP-1 analogs are proven to have a beneficial impact on hepatic health. Kahal et al. conducted a study in which they found that treatment with liraglutide, alongside associated weight loss, led to a significant reduction in procollagen type 3 amino-terminal peptide (PIIINP). This peptide serves as a predictor of liver cirrhosis, particularly in obese women with PCOS. This observation introduces an additional factor when considering the use of liraglutide in women with PCOS, obesity, and non-alcoholic fatty liver disease [63]. 

In a case-control study by Kahal et al., the effects of a 6-month management regimen with liraglutide at a daily dose of 1.8 mg was assessed on body weight, depression scale, and quality of life (QOL) in 19 young and obese women with PCOS. The results indicated a significant reduction in body weight in the PCOS group as a result of liraglutide treatment. Furthermore, a significant improvement was observed in social health, physical well-being, and psychological components of the World Health Organization QOL questionnaire (WHOQOL-BREF) [64].

In young, non-diabetic patients with PCOS, GLP-1 RAs have also provided beneficial effects on insulin sensitivity, thus addressing many manifestations of PCOS, as approximately 70% of women are insulin resistant and 80% are overweight or obese. GLP-1 RAs demonstrated significant improvements in fasting blood glucose, triglycerides, total cholesterol, and homeostasis model assessment for insulin resistance. A trial including 150 women with PCOS with impaired fasting glucose and/or impaired glucose tolerance demonstrated a prediabetes remission rate of 64% with combined treatment, 56% with exenatide only, and 32% with metformin only [65].

The weight loss effects of GLP-1 RAs offer a chance to expand the treatment options available to PCOS patients. This may be beneficial under some circumstances, such as in assisted reproductive settings when women seeking help for infertility have advanced age and/or poor ovarian reserve [18].

It was proven that a preconception treatment with low-dose liraglutide (1.2 mg daily) in combination with metformin was superior to metformin alone in increasing in vitro fertilization pregnancy rates [66]. 

In a study on rats, treatment with GLP-1 during the proestrus phase doubled the LH serum level and resulted in exerting progesterone in the luteal phase, which constantly increased the number of mature Graafian follicles, resulting in increased fertility [66].

Siamashvili et al. compared exenatide with metformin and demonstrated a significantly greater number of spontaneous pregnancies in the exenatide group after 24 weeks of treatment [67]. A review by Abdalla et al. showed that in clinical trials, exenatide, either in combination with metformin or in monotherapy, improves menstrual regularity and ovulation rate in overweight or obese women with PCOS, which directly translates to enhanced fertility [68].

However, so far, no conclusive data concerning the safety of GLP-1RAs in pregnancy are available. They were classified by the FDA and the European Medicines Agency as pregnancy class C. Therefore, women in their reproductive years should be on effective contraception while on therapy and have a washout period before trying to conceive [69].

Some data on pregnancy outcomes with exposure to liraglutide are available. Across various studies, of 111 pregnancies with known fetal outcomes, 53 (47.7%) resulted in a live birth, of which 2 (1.8%) had a congenital anomaly. Fetal loss occurred in 38 (34.2%) pregnancies due to spontaneous abortion (*n* = 32; 28.8%), ectopic pregnancy (*n* = 2; 1.8%), and stillbirth (*n* = 2; one with fetal defects). Finally, 20 (10%) pregnancies were terminated, of which 6 (5.4%) had reported fetal defects [70].

The prudent utilization of GLP-1 receptor agonists warrants thorough initial patient history screening for contraindications such as a history of pancreatitis, diabetic retinopathy, or medullary thyroid cancer. Caution is advised when administering GLP-1 analogs to patients concurrently on renin–angiotensin system inhibitors due to their heightened susceptibility to acute kidney injury resulting from dehydration and volume contraction [71]. Qualification for GLP-1RAs should align with the 5 A’s framework proposed in the clinical practice guidelines published in the Canadian Medical Association Journal in 2020. This comprehensive scheme aims to tailor treatment to a patient’s individual needs and involves steps such as raising patient awareness of obesity as a problem with metabolic consequences, seeking permission to implement treatment, setting realistic therapeutic goals, and assessing patient willingness to adhere to treatment [72]. In cases where patients have developed eating disorders or other conditions affecting their emotional relationship with food, psychological consulting may assist in advancing beneficial outcomes.

Once initiated, the administration of GLP-1 analogs necessitates vigilant monitoring for efficacy. A study conducted by Niafar et al. involving 172 patients reported that after 3 months of treatment with liraglutide, BMI dropped on average from 0.72 to 2.58 kg/m^2^. During the same period, 88 women exhibited a decrease of 0.29 nmol/L in serum testosterone levels [73].

A clinical investigation by Nylander et al. meticulously tracked ovarian changes using ultrasonography in obese PCOS patients treated with liraglutide for 6 months, which revealed a notable reduction in ovarian volume compared with patients in the placebo group. However, no improvement was observed in the Ferriman-Gallwey scale assessment of hirsutism, which may be attributed to the relatively short 6-month duration of treatment. This suggests that the influence of treatment on this particular assessment parameter might require a longer duration to achieve a substantive change [74].

In accordance with guidelines issued by the Polish Society for the Treatment of Obesity (PTLO), the therapeutic goal for weight loss in women with PCOS is a reduction in initial body weight greater than ≥5–15% of the initial body weight over a period of 3–6 months. A weekly weight reduction of 0.5 kg is deemed optimal for yielding beneficial results. Especially in the initial stages of treatment, patients are encouraged to record weekly weight measurements using a consistent scale. Regular communication with a physician is also recommended to fortify patient compliance and monitor treatment efficacy [75,76].

Patients undergoing treatment with GLP-1 agonists require close monitoring for potential adverse effects induced by the medication. GLP-1RAs do not increase the risk of hypoglycemia but exhibit a good safety profile and are generally well tolerated in PCOS management [70]. The most commonly reported adverse effects are gastrointestinal symptoms, including nausea and diarrhea, with an incidence frequency of ≥1/10. Furthermore, vomiting, constipation, abdominal pain, and dyspepsia are relatively common (with a frequency of ≥1/100 to <1/10). It is noteworthy that these symptoms are generally most intense during the early stages of treatment and tend to alleviate over time [77]. Gradual dose titration is the best strategy to lower the incidence of those reactions and increase the tolerability of the therapy [70]. Adverse symptoms at the injection site that may occur include pruritis, rash, or erythema, with pruritis being the most frequently reported injection site symptom. Injection site reactions are reported more frequently with long-acting than short-acting GLP-1 RAs. 

Currently, there are no long-term safety data for the administration of GLP-1 RAs in nondiabetic patients [66].

Clinicians must exercise caution and consider the potential risk of acute renal failure associated with the administration of GLP-1 receptor agonists. The primary mechanism by which acute kidney injury develops in people receiving GLP-1 receptor agonists is by volume contraction resulting from gastrointestinal symptoms. Consequently, patients should be advised to discontinue therapy in the event of severe vomiting or diarrhea [77].

Assessing the balance between predicted therapeutic results and possible adverse effects should always be individualized, particularly when contemplating the discontinuation of treatment.

In conclusion, when considering adding a GLP-1 analog to the metabolic treatment of a patient with PCOS, the primary indication and intended use should be to treat obesity (with a BMI > 30), or a BMI > 27, accompanied by concomitant metabolic dysfunctions such as hypertension, dyslipidemia, obstructive sleep apnea, impaired fasting glucose, impaired glucose tolerance, and type 2 diabetes.

To effectively consider a patient with PCOS for GLP-1 analog therapy, screening for metabolic syndrome is essential. This syndrome consists of impaired glucose tolerance (prediabetes), waist circumference ≥88 cm, blood pressure exceeding 130/85mmHg, high-density lipoprotein (HDL) < 1.3 mmol/L, and triglyceride (TG) levels > 1.7 mmol/L [78]. The ESHRE recommends that an oral glucose tolerance test (OGTT) be conducted in patients with PCOS with a BMI equal to or higher than 27 kg/m^2^ [2].

In cases where a patient has previously failed to attain therapeutic objectives with diet and exercise alone, the addition of a GLP-1 analog to their treatment regimen is warranted. This is especially true for patients who were treated with metformin in the past. GLP-1 analogs can be co-administered alongside metformin and other insulin sensitizers or used as monotherapy, supplanting previous therapy.

It is preferable to initiate GLP-1 analog treatment in individuals who show a willingness to comply with lifestyle changes and a structured therapeutic regimen. Prior to initiating therapy, candidates for GLP-1 treatment should be screened for exclusionary factors such as a history of hepatitis, pancreatic cancer, thyroid cancer, and psychological disorders affecting food intake.

Upon qualification and initiation of treatment, patients should initially be monitored weekly for treatment effect, particularly with respect to body weight measurements. After the initial month of treatment, the frequency of weight measurements can be reduced to once per month. Additionally, a 3-month interval timeframe is an optimal checkpoint for hormonal and lipid laboratory testing.

Facilitating a consistent line of communication between patients and physicians holds potential benefits in terms of adherence and overall success. As such, it should be actively encouraged, usually for a 6- to 12-month duration, while the treatment of metabolic impairments in PCOS is ongoing.

## Figures and Tables

**Table 1 jcm-12-05915-t001:** Comparison among glucagon-like peptide 1-receptor agonists.

Drug	Route of Administration	Frequency	Dose (mg)	Half-Life Time	Weight Loss	Cardiovascular Benefits	↓ HbA1c
Exenatide	sc	twice daily	5–10	2.4 h	++	No	++
Lixisenatide	sc	once daily	10–20	3 h	+	No	++
Liraglutide	sc	once daily	0.6–1.8	13 h	+++	Yes	++
Dulaglutide	sc	once weekly	0.75–1.5	5 days	++	Yes	++
Semaglutide	sc	once weekly	0.25–1	1 week	+++	Yes	+++
	oral	once daily	3–14	1 week	+++	Yes	+++

sc—subcutaneous. (+)—positive effect, (++)—strong positive effect, (+++)—very strong positive effect.

## Data Availability

Not applicable.

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
