# Peer review of "Do GLP-1 Analogs Have a Place in the Treatment of PCOS? New Insights and Promising Therapies"

_jcm, 2023, doi:10.3390/jcm12185915_

Round 1

Reviewer 1 Report

While there is a comprehensive discussion on the impact of metabolic syndrome on patients with PCOS and the uses and outcomes of GLP-1 agonists in the non-PCOS population, the discussion of the use of GLP-1 agonist in the PCOS population is sparse. The potential use of this medication in this particular patient population needs to be further expounded. There is also no discussion of potential risks (known or unknown) of this medication in a young, non-diabetic population - including risk of fertility, pregnancy outcomes etc. 

Author Response

Dear Reviewer, 

Thank you for your review of the article and your suggestions. 

The section concerning young patients planning to conceive was expanded in the text, including the FDA/EMA pregnancy classification of GLP-1RAs and potential effects of their continued administration. Few studies on GLP-1RAs touch on the subject of a suitable washout period, stating only that these medications are contraindicated in pregnancy. Pregnancy risks including fetal abnormalities and spontaneous abortions were added to the text.

Additionally, as it was suggested, we expanded the text concerning implementation the GLP-1 agonists in the PCOS population, referring to studies that reported beneficial effects on hepatic health (GLP-1 analogs as prevention of cirrhosis of the liver and nonalcoholic fatty liver disease), cardiometabolic condition, and atherothrombotic risk. Also, studies performed with the participation of PCOS patients were cited to show the effects on weight loss, obesity and quality of life. 

We have added new citations accordingly.

Reviewer 2 Report

Dear authors, thank you for this very well- written overview on GLP-1 analogs and PCOS. I only have minor comments: 

Abstract: please mention that exercise and change of life style is essential to achieve weight loss and was part of all studies! 

line 95: Reference is missing.Please provide which GLP1 analog was examined

line 107: Focus on PCOS, please delete paragraph on Metformin and T2DM

line 119: please add indications: BMI >27, comorbidities like... required

line 240-260: please shorten these paragraphs

line 299: Berberine-->please delete, only one study

line 326: Astrups study was not in patients with PCOS. Delete! 

line 341: As PCOS is a common disease in women of reproductive age and maybe wishing to conceive please state the current knowledge how long before conception GLP1 Analogs should be stopped. 

line 368: please cite ESHRE guideline (Teede et al., 2018) first

line 376-390: redundant to chapter 1! Delete please

Author Response

Dear Reviewer, 

Thank you for your review of the article. 

We have included the following changes to the text:

It was added to the abstract that exercise and change of lifestyle is essential to achieve weight loss and was part of all studies.

In Chapter 2, the paragraph on metformin and type 2 diabetes was removed.

Furthermore, indications for GLP-1 agonists, such as obesity and BMI>27 coinciding with metabolism-related conditions, eg. hypertension, dyslipidemia, obstructive sleep apnea were added.

Paragraphs concerning probiotic supplementation were abridged – we focused on the potential metabolic benefits of supplementation for PCOS patients.

The paragraph concerning potential use of berberine in PCOS was deleted.

The fragment referring to Astrup’s study was removed.

In line 368, ESHRE guideline (Teede et al., 2018) was cited.